# Toxicity, Response and Survival in Older Patients with Metastatic Melanoma Treated with Checkpoint Inhibitors [note 1]

**DOI:** 10.3390/cancers13112826

**Published:** 2021-06-05

**Authors:** Nienke A. de Glas, Esther Bastiaannet, Frederiek van den Bos, Simon P. Mooijaart, Astrid A. M. van der Veldt, Karlijn P. M. Suijkerbuijk, Maureen J. B. Aarts, Franchette W. P. J. van den Berkmortel, Christian U. Blank, Marye J. Boers-Sonderen, Alfonsus J. M. van den Eertwegh, Jan-Willem B. de Groot, John B. A. G. Haanen, Geke A. P. Hospers, Hilde Jalving, Djura Piersma, Rozemarijn S. van Rijn, Albert J. ten Tije, Gerard Vreugdenhil, Michel W. J. M. Wouters, Johanneke E. A. Portielje, Ellen W. Kapiteijn

**Affiliations:** 1Department of Medical Oncology, Leiden University Medical Center, 2300 RC Leiden, The Netherlands; e.bastiaannet@lumc.nl (E.B.); j.portielje@lumc.nl (J.E.A.P.); H.W.Kapiteijn@lumc.nl (E.W.K.); 2Department of Surgery, Leiden University Medical Center, 2300 RC Leiden, The Netherlands; 3Department of Gerontology and Geriatrics, Leiden University Medical Center, 2300 RC Leiden, The Netherlands; F.van_den_bos@lumc.nl (F.v.d.B.); S.P.Mooijaart@lumc.nl (S.P.M.); 4Department of Medical Oncology and Radiology and Nuclear Medicine, Erasmus MC Cancer Institute, 3015 GD Rotterdam, The Netherlands; a.vanderveldt@erasmus.nl; 5Department of Medical Oncology, University Medical Center Utrecht, 3584 CX Utrecht, The Netherlands; K.Suijkerbuijk@umcutrecht.nl; 6Department of Medical Oncology, Maastricht University Medical Center, 6229 HX Maastricht, The Netherlands; mjb.essers.aarts@mumc.nl; 7Department of Medical Oncology, Zuyderland Medical Center, 6162 BG Sittard-Geleen, The Netherlands; f.vandenberkmortel@zuyderland.nl (F.W.P.J.v.d.B.); g.a.p.hospers@umcg.nl (G.A.P.H.); m.jalving@umcg.nl (H.J.); D.Piersma@mst.nl (D.P.); 8Department of Medical Oncology, Netherlands Cancer Institute, Antoni van Leeuwenhoek Hospital, 1066 CX Amsterdam, The Netherlands; c.blank@nki.nl (C.U.B.); j.haanen@nki.nl (J.B.A.G.H.); 9Department of Medical Oncology, Radboud University Medical Center, 6500 HB Nijmegen, The Netherlands; marye.boers-sonderen@radboudmc.nl; 10Department of Medical Oncology, Cancer Center Amsterdam, Amsterdam UMC, Vrije Universiteit Amsterdam, 1105 AZ Amsterdam, The Netherlands; vandeneertwegh@amsterdamumc.nl; 11Isala Oncology Center, 8025 AB Zwolle, The Netherlands; j.w.b.de.groot@isala.nl; 12Department of Medical Oncology, University Medical Center Groningen, 9413 GZ Groningen, The Netherlands; 13Department of Medical Oncology, Medisch Spectrum Twente, 7512 KZ Enschede, The Netherlands; 14Department of Medical Oncology, Medical Center Leeuwarden, 8934 AD Leeuwarden, The Netherlands; rozemarijn.van.rijn@znb.nl; 15Department of Medical Oncology, Amphia Ziekenhuis, 4818 CK Breda, The Netherlands; atentije@amphia.nl; 16Department of Medical Oncology, 5504 DB Veldhoven, The Netherlands; G.Vreugdenhil@mmc.nl; 17Dutch Institute for Clinical Auditing, 2333 AA Leiden, The Netherlands; m.wouters@nki.nl; 18Department of Surgical Oncology, Netherlands Cancer Institute, Antoni van Leeuwenhoek Hospital, 1066 CX Amsterdam, The Netherlands; 19Department of Biomedical Data Sciences, Leiden University Medical Center, 2300 RC Leiden, The Netherlands

**Keywords:** immunotherapy, melanoma, older adults, geriatric oncology, toxicity, response

## Abstract

**Simple Summary:**

Trials suggest no differences in immunotherapy treatment between older and younger patients, but mainly young patients with a good performance status were included in these trials. The aim of this study was to describe the treatment patterns and outcomes of “real-world” older patients with metastatic melanoma. We included 2216 patients aged ≥65 years from the Dutch Melanoma Treatment Registry and described outcomes of immunotherapy. The study showed that responses and severe side effects did not differ from previously reported younger populations and randomized trials, even in the oldest patients and in patients with other diseases. However, patients aged ≥75 discontinued treatment due to toxicity more often, resulting in fewer treatment cycles. We therefore conclude that immunotherapy seems to have similar effects in older patients compared to younger patients, but the impact of less severe toxicity on quality of life should be further studied as older patients are more likely to discontinue treatment.

**Abstract:**

Background: Previous trials suggest no differences in immunotherapy treatment between older and younger patients, but mainly young patients with a good performance status were included. The aim of this study was to describe the treatment patterns and outcomes of “real-world” older patients with metastatic melanoma and to identify predictors of outcome. Methods: We included patients aged ≥65 years with metastatic melanoma from the Dutch Melanoma Treatment Registry. We described the reasons for hospital admissions and treatment discontinuation. Additionally, we assessed predictors of toxicity and response using logistic regression models and survival using Cox regression models. Results: We included 2216 patients. Grade ≥3 toxicity was not associated with age, comorbidities or WHO status. Patients aged ≥75 discontinued treatment due to toxicity more often, resulting in fewer treatment cycles. Response rates were similar to previous trials (40.3% and 43.6% in patients aged 65–75 and ≥75, respectively, for anti-PD1 treatment) and did not decrease with age or comorbidity. Melanoma-specific survival was not affected by age or comorbidity. Conclusion: Response rates and toxicity outcomes of checkpoint inhibitors did not change with increasing age or comorbidity. However, the impact of grade I-II toxicity on quality of life deserves further study as older patients discontinue treatment more frequently.

## 1. Introduction

Immune checkpoint inhibition has strongly improved the outcome of patients with advanced and metastatic melanoma in the past decade [1,2,3]. However, only around 40% of patients respond to immunotherapy, and so far, it remains difficult to predict which patients will respond.

Currently, 43% of patients with melanoma in the Netherlands are aged 65 years or older, and this percentage will increase strongly in the upcoming decades due to the aging population [4]. Randomized studies have suggested no strong differences in response rates or toxicity between younger and older patients with melanoma [5]. However, older patients were systematically underrepresented in randomized clinical trials in which checkpoint inhibitors were studied, and only “fit” patients without comorbidities and a WHO status of 0 or 1 were included [5]. This means that the results of these trials cannot be directly extrapolated to the general older population who frequently have multiple comorbidities and a worse performance status.

The challenge of treating cancer in older adults lies in the large heterogeneity of this population. While some older patients are generally fit and are able to tolerate anticancer treatments without major problems, others have concomitant diseases, cognitive impairment, physical deficits and/or a poor social support system [6]. In general, patients with these factors are underrepresented and underreported in randomized clinical trials [7]. Unfortunately, none of the randomized phase III trials in immunotherapy included any measures of frailty or biological age in their reports [5,8]. Therefore, it is essential to study outcomes of immunotherapy in real-life older populations.

Hence, the aim of this study was to describe treatment patterns and outcomes of older patients with metastatic melanoma in real-world data, and to identify predictors of outcome, using data from the nationwide Dutch Melanoma Treatment Registry (DMTR).

## 2. Materials and Methods

### 2.1. Patient Inclusion

In the Netherlands (with 17 million inhabitants), systemic treatment for patients with melanoma is centralized in 14 hospitals. Data from all patients treated in these hospitals are included in the DMTR, irrespective of treatment strategy [9]. In this national treatment registry, detailed information on tumor characteristics, patient characteristics, treatments, response rates, survival and toxicity outcomes (according to the NCI-CTC criteria [10]) are registered by trained data managers. The prospective registry started in 2013 and includes all patients with unresectable stage III or stage IV melanoma. Patients who did not undergo treatment after consultation with a melanoma treatment center, and patients who did visit a melanoma treatment center but did not receive treatment, are also included in the registry. However, patients for whom consultation with a melanoma treatment center did not occur are not included in the registry.

In this study, we included all patients aged 65 years or older diagnosed with metastatic (stage IV) melanoma between July 2013 and March 2020. Patients with a primary uveal melanoma were excluded. The study design was approved by the scientific board of the DMTR.

### 2.2. Variables

Patient characteristics that were tested included age, number of comorbid diseases, autoimmune disease (for toxicity outcomes), and WHO status. With regard to comorbidity, the DMTR includes a predefined list of relevant comorbidities that can be checked (or unchecked) per patient, as has been previously published [9]. We counted the total number of registered comorbidities. No formal comorbidity score (such as the Charlson or CIRG-score) has been included in the DMTR, but we have previously shown that a comorbidity count is just as good in predicting the outcome of older adults [11].

Tumor characteristics included disease stage, number of metastatic sites, histology type (superficial, nodular, acrolentiginous, lentigo maligna, desmoplastic, other or unknown), brain metastases at baseline and tumor mutations (including BRAF, NRAS and KIT). Other prognostic factors included baseline LDH (with a cut-off of 250 mmol/L).

Treatments that were studied included anti-PD1 therapy (nivolumab, pembrolizumab), anti-CTLA-4 therapy (ipilimumab) or a combination of these two during any line of treatment.

### 2.3. Statistical Analyses

All analyses were performed in SPSS version 24.0. Values of *p =* 0.05 or smaller were considered statistically significant, and all tests were two-sided.

In order to describe treatment patterns in different age groups, we divided patients into two groups: Patients aged 65–74 and patients aged 75 years or older. First, we compared baseline tumor, patient and treatment characteristics between these groups using Chi-square tests in order to assess whether these factors may change with increasing age.

Because we know from previous studies [12] that immunotherapy treatment patterns have changed in recent years, we depicted first-line treatments graphically per age group, as defined above.

Next, we assessed predictors of grade ≥3 toxicity using logistic regression models, per treatment strategy (either anti-PD1 treatment, ipilimumab or combination). All patient characteristics were first analyzed in the univariate analyses. Toxicity during any line of treatment was included for this analysis. Additionally, we reported reasons for discontinuation and hospital admissions, comparing the two age groups using descriptive statistics including Chi-square tests for categorical outcomes and unpaired *t*-tests for continuous outcomes.

We assessed predictors of treatment response of the first-line treatment after 6 months using logistic regression models. Treatment response was defined as either a complete response or partial response. Patients who died before the first evaluation scan were classified as nonresponders. Due to the small number of older patients receiving combination treatment (ipilimumab + nivolumab), we were only able to reliably analyze predictors of response to anti-PD1 monotherapy or ipilimumab. We performed logistic regression analyses in which tumor and patient characteristics were entered as independent variables and treatment response within 6 months as a dependent variable.

Finally, we studied predictors of melanoma-specific survival and overall survival using univariate and multivariate Cox regression models. Again, all clinically relevant and significant predictors were included in the multivariate model.

### 2.4. Sensitivity Analyses

In order to further investigate reasons for treatment discontinuation, we compared patients who discontinued treatment due to toxicity or due to other reasons using Chi-square statistics. Furthermore, we investigated whether the number of treatment cycles differed between younger and older patients.

## 3. Results

### 3.1. Patient Characteristics

Overall, 2216 patients were included. Patients characteristics are presented in Table 1. A total of 1139 patients aged 65–74 and 1077 aged ≥75 were included. The majority of patients were male (64.7% versus 35.3%). Most patients had one or more comorbidities (43.3% had 1–2 comorbidities and 40.5% had 3 or more comorbidities), and this increased with age (*p* < 0.001). Older patients more frequently presented with a worse performance score at baseline.

Forty percent of patients had a BRAF-mutated tumor. BRAF was less frequently tested in the oldest patients, resulting in more unknowns in the oldest age group (5.2% in patients aged 65–74 and 9.0% in patients aged ≥75, *p* < 0.001). LDH was elevated in 37.3% of all patients, and, again, this was more frequently unknown in the oldest age group (10% versus 7.4, *p* < 0.001). Brain imaging (using CT or MRI) was performed less frequently in the oldest age group (65.4% in patients aged 65–74 versus 59.1% in patients aged 75 years or older, *p* = 0.037).

Ipilimumab (during any line of treatment) was prescribed in 235 patients aged 65–74 (20.6%) and 129 patients aged 75 years or older (12.0%, *p* < 0.001). Anti-PD1 was prescribed in 41.5% of all patients (with no differences between age groups (*p* = 0.222)). Combination immunotherapy was prescribed in 12.6% of patients aged 65–74 and 5.3% of patients aged 75 years or older (*p* < 0.001). In patients aged ≥75, significantly more patients received best supportive care after diagnosis (13.9% versus 6.1% in patients <75, *p* < 0.001). 

First-line treatments changed over time in both age groups (Figure 1). While, in 2013, the majority of patients were treated with a BRAF inhibitor or chemotherapy, after 2017, most patients were treated with anti-PD1 monotherapy as first-line treatment.

### 3.2. Treatment Toxicity

The risk of treatment-related toxicity did not increase with increasing age (Table 2). Anti-PD1 toxicity (grade ≥3) occurred in 13.9% of patients aged 65–74 and 16.6% of patients aged 75 years or older (*p* = 0.255). The most frequently reported toxicities included colitis (2.8%), hepatitis (2.2%) and pneumonitis (1.7%). Ipilimumab toxicity (grade 3 or higher) occurred in 31.9% of all patients without differences between age groups. The most frequent toxicities included colitis (13.7%) and hypopituitary insufficiency (6.3%, Appendix A). For combination treatment, 41.0% of patients aged 65–74, and 47.4% of patients aged ≥75 had grade ≥3 toxicity (*p* = 0.0543), of which the most frequent toxicities included colitis (12.9%) and hepatitis (10.4%).

There were no statistically significant predictors of immunotherapy toxicity for either ipilimumab, anti-PD1 treatment or combination treatment (including age, sex, number of comorbidities, autoimmune diseases or performance status, Table 2).

In patients treated with anti-PD1, the oldest patients received significantly fewer treatment cycles compared to younger patients (*p* < 0.001). For ipilimumab, no differences in the number of treatment cycles were observed. For combination treatment, only 32.8% of all patients completed their 4 treatment cycles, and this deteriorated with increasing age (38.9% in patients aged 65–74 and 17.5% of patients aged ≥75, *p* = 0.003). The most frequent reasons for discontinuation were progression, planned ending of treatment and treatment toxicity (Appendix A). Older patients (>75) discontinued anti-PD1 treatment due to toxicity more frequently than younger patients (17.6% versus 12.2%, respectively, *p* = 0.026), but this difference was not statistically significant for ipilimumab or combination treatment. Patients had unplanned hospital admissions on average 3.5 times, and this did not differ between the two age groups (*p* = 0.263). The most frequent reasons for hospital admissions included supportive care (29.8%) and treatment for toxicity (24.2%) (Appendix A). As sensitivity analyses, we investigated differences between patients who discontinued anti-PD1 treatment due to toxicity versus patients who discontinued treatment due to other reasons (Appendix A). Patients who discontinued anti-PD1 treatment due to toxicity were older (41.2% were 65–74 and 58.8% ≥75 compared to 53.5% of patients aged 65–74 and 46.5% aged ≥75 in patients who discontinued treatment due to other reasons, *p* = 0.006). There were no differences in comorbidities or baseline WHO status. Thirty-six percent of patients who discontinued treatment due to toxicity had a response within six months.

### 3.3. Treatment Response

Response percentages and predictors of response to first-line treatment after six months are presented in Table 3. A total number of 681 patients received anti-PD1 treatment as first-line treatment, while 207 patients received ipilimumab as first-line treatment. Age, sex and comorbidities were not associated with the probability of treatment response for both anti-PD1 and ipilimumab. In patients treated with anti-PD1 treatment, a poor performance status was associated with a lower probability of response (OR: 0.54, 95 confidence interval (C.I.) 0.38–0.76 for WHO 1, OR: 0.58, 95% C.I.: 0.32–1.04 for WHO 2 and OR: 0.35, 95% C.I.: 0.07–1.175 for WHO 3/4, *p* = 0.006). For ipilimumab monotherapy, similar point estimates were observed, but these were not statistically significant. An increased LDH level at baseline was predictive of a lower probability of response for both anti-PD1 treatment (OR: 0.68, 95% C.I.: 0.48–0.96) and ipilimumab (OR: 0.20, 95% C.I.: 0.05–0.85). The presence of brain metastases was not predictive for treatment response for both treatments.

### 3.4. Survival Outcomes

Median follow-up was 0.7 years (range: 0–7.2 years) for the whole cohort. Melanoma-specific survival analyses are presented in Table 4. In the univariate analyses, age, sex and comorbidities were not associated with melanoma-specific survival. A poor performance status at baseline was associated with poor melanoma-specific survival (HR: 1.73, 95% C.I.: 1.49–2.00 for WHO 1, HR: 2.74, 95% C.I.: 2.26–3.33 for WHO 2, and HR: 5.30, 95% C.I.: 4.12–6.83 for WHO 3–4 compared to WHO 0, *p* < 0.001), and this remained statistically significant in the multivariate analyses. Additionally, the number of metastatic sites, LDH level at baseline and the presence of brain metastases at baseline were poor prognostic factors for melanoma-specific survival.

For overall survival, the main predictors in the univariate analyses included age (HR: 1.22, 95% C.I.: 1.1–1.35, *p* < 0.001 for patients aged 75 years or older versus patients aged 65–74, Appendix A), number of comorbidities (*p* = 0.005), WHO status (*p* < 0.001), LDH level (*p* < 0.001) and brain metastases (*p* < 0.001). In the multivariable analyses, point estimates remained similar to univariate analyses, but only age, WHO status, LDH and the presence of brain metastases remained statistically significant.

## 4. Discussion

This study showed that treatment toxicity and response rates in older patients with metastatic melanoma in a real population were comparable to those reported in previous randomized clinical trials and previous real-world data in younger patients [2,13,14,15]. Grade ≥3 toxicity did not increase in the oldest patients or in patients with multiple comorbidities, but the oldest patients did discontinue treatment due to toxicity more frequently and received fewer treatment cycles. Combination treatment with ipilimumab + nivolumab was completed in only 32.8% of all patients. A poor performance status at baseline was the most important predictor for treatment response and melanoma-specific survival.

Because our study showed a similar response and toxicity rates to previous studies, it confirms that there are no large differences in response and toxicity rates between younger and older patients and in patients who were currently selected for treatment with immunotherapy [5,12,15,16,17]. These results are important, as a previous study using DMTR data showed that older patients are less frequently treated with second-line immunotherapy compared to patients under 65 (submitted), which may not be justified based on our findings. However, for combination treatment, only one-third of older patients completed the treatment, and grade ≥3 toxicity was higher in the oldest age group (although not statistically significant).

In addition, older patients received a less complete diagnostic workup with less BRAF testing and less imaging of the brain, which may result in undertreatment. Additionally, we showed that known predictors of response and melanoma-specific survival including brain metastases, LDH and a high tumor burden [18,19] were important predictors in older patients and may be used to guide treatment decisions in this age group as well. WHO performance status was strongly associated with response and survival, which was most likely explained by the fact that these patients already had a large disease burden and were less able to undergo intensive treatments, but not by aging-related decline.

Interestingly, some small observational studies have investigated outcomes of immunotherapy in older patients. These studies did not show age differences with regard to grade ≥3 toxicity and response rates, but a higher rate of early treatment discontinuation and grade I-II adverse events was observed in older patients, especially in patients with multiple comorbidities or a poor performance status at baseline [8,17]. In addition, an overview of previous phase I trials in melanoma showed an increased incidence of grade I-II adverse events in older patients [20], and this was recently confirmed in a large observational study in >600 older patients on anti-PD1 treatment [21].

Similarly, we observed that the oldest patients (≥75) received fewer treatment cycles, and discontinued anti-PD1 treatment more frequently due to toxicity than patients aged 65–74, probably explained by grade I-II adverse events, as we did not observe differences in grade ≥3 toxicity. In older, especially frail patients, grade I-II toxicities may not only be relevant because of the risk of early treatment discontinuation, but it can also lead to morbidity and deterioration of quality of life and physical functioning [8], especially when long lasting. In addition, it is possible that the oncological toxicity scoring does not adequately reflect side effects that are typical and relevant in older patients, such as functional or cognitive decline. This may be an explanation why we found more treatment discontinuation in this age group.

Unfortunately, the DMTR does not include any measurements of frailty or geriatric assessment. It must be noted that the WHO performance status in older patients is not very reliable, as oncologists frequently underestimate the physical deficits that patients have [6,22,23]. This might explain why the performance status did not predict toxicity in this population. In patients treated with chemotherapy, previous studies have demonstrated that factors associated with aging (such as cognitive impairments, poor nutritional status and polypharmacy) were strongly associated with toxicity and survival and were more predictive than age and performance status alone [24,25,26,27]. Similarly, a recent study in older patients who were treated with checkpoint inhibitors showed that 60% of patients with a low score at the Geriatric-8 (G8), were scored as performance status of 0 or 1, and this was predictive for toxicity and early death [28]. Future studies should therefore include both geriatric assessment at baseline and relevant outcomes for these older patients, including quality of life and physical and cognitive functioning.

### Strengths and Limitations

The major strength of this study lies in the large sample size in which all patients in the Netherlands with metastatic melanoma for whom consultation occurred in a melanoma treatment center were included. The DMTR provides well-registered, validated and detailed information in a large real-life population. To our knowledge, this is the largest study in real-life data in older patients treated with checkpoint inhibitors. Furthermore, we were able to study not only predictors of response, survival and toxicity, but also treatment discontinuation and hospitalizations, which gave more insight into treatment outcomes in this older population.

However, this study has some limitations. First, we were only able to study grade ≥3 toxicity. Grade I-II toxicity may also be important to older, especially frail patients, as it may lead to a decrease in quality of life and treatment discontinuation. We were only able to assess the predictive capacity of age, comorbidities and performance status as aging-related predictors. No geriatric assessment data were available, while these are essential to estimate characteristics of frailty in an older population [6]. Additionally, we were not able to analyze outcomes such as quality of life or physical functioning. The DMTR will include patient-reported outcomes in the future, which will be especially relevant for the older population [9]. Furthermore, we had a substantial amount of missing data for some of the variables (such as WHO status, number of metastases and tumor mutations). In addition, follow-up for survival outcomes was relatively short, as we decided to include as many patients as possible (including the most recent years). This was however not an issue for the main outcomes of the study including response and toxicity. Finally, patients who decided with their primary care physician to omit treatment without consultation of a melanoma treatment center are not included in the DMTR, which means that this might be a partly selected population. This is reflected by the fact that there were relatively few patients with multiple comorbidities or a performance status of three or four. Still, our patient cohort was much older and had a worse performance status than those included in the previous phase 3 immunotherapy trials. For example, patients in the Keynote-006 trial had a median age of 61, and no patients with a performance status of 2 or higher were included [29].

## 5. Conclusions

In conclusion, response rates and toxicity outcomes of checkpoint inhibitors did not change with increasing age or comorbidity. However, the impact of grade I-II toxicity in older patients deserves further study as older patients discontinue treatment more frequently and receive fewer treatment cycles. In order to further individualize treatment, future studies should investigate aging-related predictors and quality of life in older patients treated with checkpoint inhibitors.

## Figures and Tables

**Figure 1 cancers-13-02826-f001:**
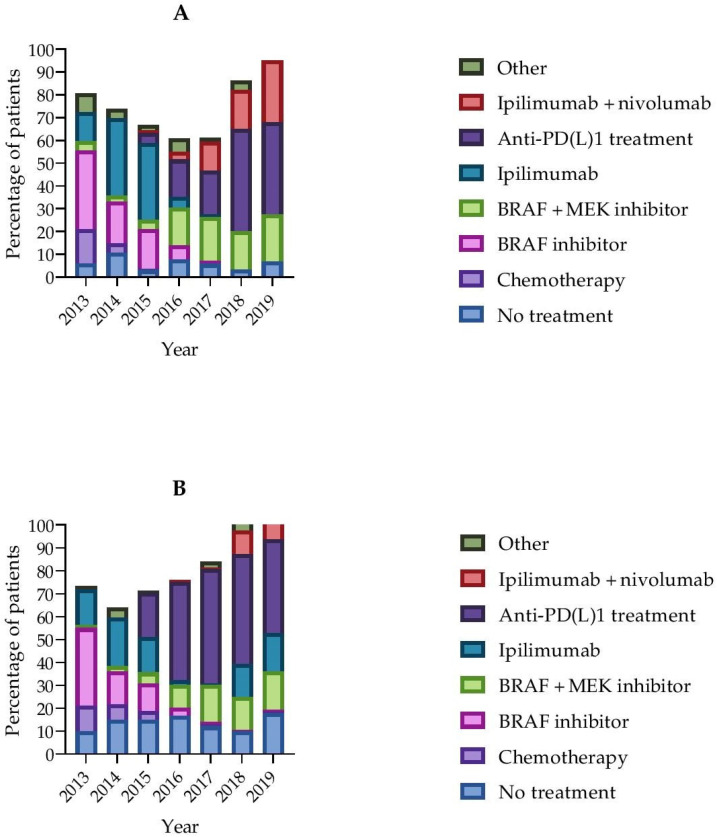
Changes in treatment strategies over time per age group: (**A**) patients aged 65–74; (**B**) patients aged 75 years and older.

**Table 1 cancers-13-02826-t001:** Patient characteristics.

	All Patients (*n* = 2216)		Patients Aged 65–74 (*n* = 1139)		Patients Aged ≥ 75 (*n* = 1077)		
	N	(%)	N	(%)	N	(%)	*p*-Value
**Patient characteristics**							
Sex							
Male	1434	(64.7)	752	(66.0)	682	(63.3)	*0.184*
Female	782	(35.3)	387	(34.0)	395	(36.7)	
Number of comorbidities							
0	309	(13.9)	215	(18.9)	94	(8.7)	*<0.001*
1–2	959	(43.3)	525	(46.1)	434	(40.3)	
3 or more	897	(40.5)	365	(32.0)	532	(49.4)	
Unknown	51	(2.3)	34	(3.0)	17	(1.6)	
WHO classification							
0	818	(36.9)	485	(42.6)	333	(30.9)	*<0.001*
1	717	(32.4)	332	(29.1)	385	(35.7)	
2	271	(12.2)	124	(10.9)	147	(13.6)	
3 or 4	118	(5.3)	57	(5.0)	61	(5.7)	
Unknown	292	(13.2)	141	(12.4)	151	(14.0)	
**Tumor characteristics**							
Number of metastatic sites							
1–2	267	(12.0)	133	(11.7)	134	(12.4)	*0.628*
3–5	195	(8.8)	104	(9.1)	91	(8.4)	
6 or more	1386	(62.5)	722	(63.3)	664	(61.6)	
Unknown	370	(16.7)	181	(15.9)	189	(17.5)	
Brain metastases							
No	1572	(70.9)	787	(69.1)	785	(72.9)	*0.016*
Yes	578	(26.1)	324	(28.4)	254	(23.6)	
Unknown	66	(3.0)	28	(2.5)	38	(3.5)	
Brain imaging (CT or MRI) at baseline							
No	790	(35.6)	381	(33.5)	409	(38.0)	*0.037*
Yes	1368	(61.7)	732	(64.3)	636	(59.1)	
Unknown	58	(2.6)	26	(2.3)	32	(3.0)	
Mutation status *							
BRAF							
Negative	1161	(52.4)	461	(49.3)	600	(55.7)	*<0.001*
Positive	899	(40.6)	519	(45.6)	380	(35.3)	
Unknown	156	(7.0)	59	(5.2)	97	(9.0)	
LDH							
Normal	1224	(55.2)	647	(56.8)	577	(53.6)	*<0.001*
Elevated	827	(37.3)	435	(38.2)	392	(36.4)	
Unknown	165	(7.4)	57	(5.0)	108	(10.0)	
Type of treatment **							
BRAF inhibition (monotherapy)	256	(11.6)	158	(13.9)	98	(9.1)	*<0.001*
BRAF and MEK inhibition	375	(16.9)	235	(20.6)	140	(13.0)	*<0.001*
Ipilimumab	364	(16.4)	235	(20.6)	129	(12.0)	*<0.001*
Anti-PD(L)1	920	(41.5)	481	(42.2)	439	(40.8)	*0.483*
Ipilimumab + nivolumab	201	(9.0)	144	(12.6)	57	(5.3)	*<0.001*
Best supportive care after diagnosis							
No	1998	(90.1)	1070	(93.9)	928	(86.1)	*<0.001*
Yes	220	(9.9)	70	(6.1)	150	(13.9)	

* Mutation present in any of the performed biopsies during follow-up. ** Described therapy in any line of treatment. Some patients were treated with more than one line of immunotherapy.

**Table 2 cancers-13-02826-t002:** Determinants of grade ≥3 immunotherapy toxicity.

	Anti-PD(L)1				Ipilimumab				Ipilimumab + Nivolumab			
	% of Treated Patients with Toxicity	OR	95% C.I.	*p*-Value	% of Treated Patients with Toxicity	OR	95% C.I.	*p*-Value	% of Treated Patients with Toxicity	OR	95% C.I.	*p*-Value
Age												
65–74	13.9	Ref		*0.255*	31.9	Ref		*0.859*	41.0	Ref		*0.543*
75+	16.6	1.23	(0.86–1.77)		31.0	0.96	(0.60–1.52)		47.4	1.02	(0.96–1.09)	
Sex												
Male	15.4	Ref		*0.962*	32.6	Ref		*0.554*	45.0	Ref		*0.337*
Female	14.9	0.96	(0.66–1.41)		29.6	0.87	(0.54–1.39)		37.7	0.74	(0.40–1.37)	
Number of comorbidities												
0	12.1	Ref		*0.781*	28.6	Ref		*0.922*	43.9	Ref		*0.410*
1–2	15.3	1.32	(0.71–2.45)		32.7	1.22	(0.67–2.20)		46.7	1.12	(0.53–2.35)	
3 or more	16.0	1.39	(0.75–2.60)		32.8	1.22	(0.65–2.28)		34.4	0.67	(0.30–1.51)	
Unknown	15.8	1.37	(0.35–5.29)		0.0	.			55.6	1.60	(0.37–6.83)	
Auto-immune disease												
No	15.4	Ref		*0.570*	31.2	Ref		*0.521*	42.9	Ref		*0.936*
Yes	13.0	0.82	(0.41–1.63)		37.5	1.33	(0.56–3.12)		41.7	0.95	(0.29–3.11)	
WHO classification												
0	15.2	Ref		*0.480*	34.3	Ref		*0.321*	47.8	Ref		*0.704*
1	15.1	0.99	(0.66–1.48)		25.9	0.67	(0.40–1.12)		40.5	0.75	(0.40–1.40)	
2	22.4	1.61	(0.89–2.93)		50.0	1.91	(0.65–5.68)		45.0	0.89	(0.34–2.37)	
3 or 4	0.0				0.0							
Unknown	11.8	0.75	(0.34–1.63)		27.8	0.74	(0.34–1.61)		28.6	0.44	(0.13–1.50)	

**Table 3 cancers-13-02826-t003:** Determinants of response to first-line treatment after 6 months.

	Anti-PD(L)1 Treatment as First-Line Treatment (*n* = 681)				Ipilimumab as First-Line Treatment (*n* = 207)			
	6-Month Response Rate (%)	HR	(95% C.I.)	*p*-Value	6-Month Response Rate (%)	HR (95% C.I.)	*p*-Value
Age							
65–74	40.3	Ref		*0.377*	15.3	Ref	*0.842*
75+	43.6	1.15	(0.85–1.56)		14.3	0.92	*(0.41–2.08)*
Sex							
Male	42.3	Ref		*0.835*	14.2	Ref	*0.641*
Female	41.5	0.97	(0.70–1.33)		16.7	1.21	*(0.54–2.70)*
Number of comorbidities							
0	46.5	Ref		*0.506*	23.9	Ref	*0.223*
1–2	38.9	0.73	(0.44–1.23)		10.4	0.37	*(0.14–0.95)*
3 or more	44.1	0.91	(0.54–1.53)		14.1	0.52	*(0.20–1.38)*
Unknown	42.1	0.84	(0.30–2.33)		100.0	.	
WHO classification							
0	49.0	Ref		*0.006*	16.7	Ref	*0.511*
1	34.0	0.54	(0.38–0.76)		9.7	0.54	*(0.20–1.41)*
2	35.7	0.58	(0.32–1.04)		14.3	0.83	*(0.10–7.30)*
3 or 4	25.0	0.35	(0.07–1.75)		0.0		
Unknown	42.2	0.76	(0.41–1.43)		22.2	1.43	*(0.43–4.79)*
**Tumor characteristics**							
Number of metastatic sites							
1–2	39.1	Ref		*0.288*	13.3	Ref	*0.971*
3–5	44.1	1.23	(0.61–2.48)		16.7	1.30	*(0.21–8.15)*
6 or more	39.5	1.02	(0.60–1.72)		14.4	1.09	*(0.23–5.21)*
Unknown	48.0	1.43	(0.81–2.53)		17.2	1.35	*(0.23–7.98)*
LDH							
Normal	44.8	Ref		*0.067*	17.9	Ref	*0.066*
Elevated	35.6	0.68	(0.48–0.96)		4.1	0.20	*(0.05–0.85)*
Unknown	37.5	0.74	(0.26–2.07)		28.6	1.84	*(0.34–9.97)*
Brain metastases							
No	42.9	Ref		*0.224*	17.4	Ref	*0.259*
Yes	35.5	0.73	(0.48–1.12)		5.7	0.29	*(0.07–1.27)*
Unknown	52.6	1.48	(0.59–3.69)		0.0	.	
Mutation status *							
BRAF							
Negative	40.9	Ref		*0.206*	11.2	Ref	*0.081*
Positive	46.3	1.25	(0.89–1.75)		23.6	2.46	*(1.09–5.53)*
Unknown	28.6	0.58	(0.22–1.52)		22.2	2.27	*(0.43–11.87)*

* Response was defined as a partial response or a complete response.

**Table 4 cancers-13-02826-t004:** Determinants of melanoma-specific survival.

	Univariate			Multivariable		
	HR	(95% C.I.)	*p*-Value	HR	(95% C.I.)	*p*-Value
Age						
65–74	Ref		*0.531*	Ref		*0.472*
75+	1.04	(0.92–1.17)		1.05	(0.92–1.19)	
Sex						
Male	Ref		*0.611*	Ref		*0.0.761*
Female	0.97	(0.85–1.10)		0.98	(0.86–1.12)	
Number of comorbidities						
0	Ref		*0.257*	Ref		*0.606*
1–2	1.06	(0.88–1.28)		1.09	(0.90–1.31)	
3 or more	1.18	(0.98–1.42)		1.09	(0.90–1.32)	
Unknown	1.12	(0.72–1.75)		1.33	(0.85–2.10)	
WHO classification						
0	Ref		*<0.001*	Ref		*<0.001*
1	1.73	(1.49–2.00)		1.54	(1.32–1.80)	
2	2.74	(2.26–3.33)		2.24	(1.83–2.74)	
3 or 4	5.30	(4.12–6.83)		3.75	(2.88–4.88)	
Unknown	1.76	(1.44–2.15)		1.82	(1.48–2.25)	
**Tumor characteristics**						
Number of metastatic sites						
1–2	Ref		*<0.001*	Ref		*0.047*
3–5	1.27	(0.94–1.72)		1.36	(1.00–1.85)	
6 or more	2.48	(1.99–3.10)		2.28	(1.81–2.87)	
Unknown	1.36	(1.04–1.78)		1.39	(1.05–1.84)	
LDH						
Normal	Ref		*<0.001*	Ref		*<0.001*
Elevated	2.21	(1.95–2.51)		1.73	(1.52–1.97)	
Unknown	1.30	(1.02–1.65)		1.14	(0.88–1.46)	
Brain metastases						
No	Ref		*<0.001*	Ref		*<0.001*
Yes	2.10	(1.85–2.40)		1.84	(1.60–2.10)	
Unknown	1.85	(1.32–2.59)		1.80	(1.28–2.54)	

## Data Availability

The data underlying this article were provided by the Dutch Meanoma Treatment Registry (DMTR) by permission. Data will be shared on request to the corresponding author with permission of the scientific board of the DMTR.

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
