# Peer review of "Toxicity, Response and Survival in Older Patients with Metastatic Melanoma Treated with Checkpoint Inhibitors"

_cancers, 2021, doi:10.3390/cancers13112826_

Round 1

Reviewer 1 Report

The manuscript "Toxicity, Response and Survival in Older Patients with Meta-2 static Melanoma Treated with Checkpoint Inhibitors" discusses an important and interesting topic mainly relevant to clinical dermatooncologists. The study is well designed, results clearly presented and discussion well-structured. The study has some limitations, also presented by the authors themselves. Here further studies on this topic would be interesting for the future. The quality of the figures seems a bit decreased. It should be checked whether the resolution could be increased. 

Author Response

We thank the reviewer for his/her kind words. Concerning the quality of the figures, we hereby upload high-quality figures that will hopefully be better readable.  

Reviewer 2 Report

This is an excellent study of a cohort of older patients with metastatic melanoma who were treated by checkpoint inhibitors. The overall conclusion is that roughly these drugs are as active, reasonably as well tolerated than in younger patients (less than 65 years old). However, there are slight differences in tolerance and reasons to stop treatment according to age, PS and comorbidities.

There is few to comments in the general presentation of data. I presume that toxicities are reported according to NCI-CTC criteria. It should be mentioned. Introduction, materials and methods, results are clearly described. Personally, I believe that table 4 should be “determinants of overall survival”, because this is really the final endpoint of treatment, particularly in the older patients (who can die of cancer but also of other cause- disease).  

In the discussion, the statement “This study showed that treatment toxicity and response rates in older patients with metastatic melanoma in the real population were comparable to those reported in previous randomized clinical trials and previous real-world data in younger patients” should be referenced (results of the most important prospective trials.

Otherwise, the discussion is focused on several aspects of geriatric oncology. I agree with the observation that patients older than 75 years had less diagnostic/prognostic procedures and stopped treatment earlier than younger patients, which may be due to grade1-2 toxicities. These toxicities are likely to be poorly tolerated and to induce worse quality of life. Nevertheless, I have some inaccuracies to highlight.

  • Line 298: G8 score is not a tool to diagnose frailty. G8 score is only a screening score of frailty. That means that patients with G8 score < 15/17 should receive Multidimensional Geriatric Evaluation (MGE) to determine whether they are vulnerable/frail/too sick. MGE is an evaluation of health status which allows to determine what are the factors of frailty: functional, nutritional, cognitive, thymic, social, etc.
  • In this study only three criteria to evaluate health status are accessible: age, PS and comorbidities.
    1. Age is not really a good factor of evaluation (apart the limit 85 years).
    2. The authors point out the fact that 1- PS is often underestimated by physicians, this is true; 2- that few patients have a PS 3-4.
    3. But more importantly the authors do not indicate how comorbidities are evaluated. There are generally two options: to use 1- Charlson score (good tool in the kind of study of this manuscript); 2- CIRS-G (better tool to use in the practice because comorbidities are weighted. However, these tools were not available in this data base, but the authors should explain how the co-morbidities were counted.
    4. The authors should recommend more precisely some extend of the data base to allow to better analyze the older patients outcome

Reviewer 3 Report

Authors have presented a concise report on the patients 65 and above. While the information is not entirely novel, the findings support use of checkpoint blockade in this age group. My minor comment is that authors format Table 2 so that the numbers show in one row.

Author Response

We thank the reviewer for his/her comments. We have formatted the table at page 7.